# Neonatal Mesenchymal Stem Cell Treatment Improves Myelination Impaired by Global Perinatal Asphyxia in Rats

**DOI:** 10.3390/ijms22063275

**Published:** 2021-03-23

**Authors:** Andrea Tapia-Bustos, Carolyne Lespay-Rebolledo, Valentina Vío, Ronald Pérez-Lobos, Emmanuel Casanova-Ortiz, Fernando Ezquer, Mario Herrera-Marschitz, Paola Morales

**Affiliations:** 1Molecular & Clinical Pharmacology Program, ICBM, Faculty of Medicine, University of Chile, Santiago 8380453, Chile; ac.tapiabustos@gmail.com (A.T.-B.); carolynelespay@gmail.com (C.L.-R.); valeviomunoz@gmail.com (V.V.); ronald.perezlobos@gmail.com (R.P.-L.); emmanuel.casanovao@gmail.com (E.C.-O.); 2Faculty of Medicine, School of Pharmacy, Universidad Andres Bello, Santiago 8370149, Chile; 3Centro de Medicina Regenerativa, Facultad de Medicina Clínica Alemana, Universidad del Desarrollo, Av. Las Condes 12438, Lo Barnechea, Santiago 7710162, Chile; eezquer@udd.cl; 4Department of Neuroscience, Faculty of Medicine, University of Chile, Santiago 8380453, Chile

**Keywords:** hypomyelination, oligodendrocyte, myelination, neonatal asphyxia/ischemia, apoptosis, mesenchymal stem cells, telencephalon, rat brain, neuroinflammation, periventricular leukomalacia

## Abstract

The effect of perinatal asphyxia (PA) on oligodendrocyte (OL), neuroinflammation, and cell viability was evaluated in telencephalon of rats at postnatal day (P)1, 7, and 14, a period characterized by a spur of neuronal networking, evaluating the effect of mesenchymal stem cell (MSCs)-treatment. The issue was investigated with a rat model of global PA, mimicking a clinical risk occurring under labor. PA was induced by immersing fetus-containing uterine horns into a water bath for 21 min (AS), using sibling-caesarean-delivered fetuses (CS) as controls. Two hours after delivery, AS and CS neonates were injected with either 5 μL of vehicle (10% plasma) or 5 × 10^4^ MSCs into the lateral ventricle. Samples were assayed for myelin-basic protein (MBP) levels; Olig-1/Olig-2 transcriptional factors; Gglial phenotype; neuroinflammation, and delayed cell death. The main effects were observed at P7, including: (i) A decrease of MBP-immunoreactivity in external capsule, corpus callosum, cingulum, but not in fimbriae of hippocampus; (ii) an increase of Olig-1-mRNA levels; (iii) an increase of IL-6-mRNA, but not in protein levels; (iv) an increase in cell death, including OLs; and (v) MSCs treatment prevented the effect of PA on myelination, OLs number, and cell death. The present findings show that PA induces regional- and developmental-dependent changes on myelination and OLs maturation. Neonatal MSCs treatment improves survival of mature OLs and myelination in telencephalic white matter.

## 1. Introduction

Perinatal asphyxia (PA), a prototype of obstetric complication, is the result of impaired gas exchange during labor or delivery, a major cause of death, neuropsychiatric dysfunctions, and learning disorders, short and long term, in affected newborns [1], not only increasing the costs associated with the acute treatment, but also comorbidities in the life of the individuals. The duration of the perinatal insult, a key factor for the severity of the clinical outcome [2], as well as the developmental stage of the affected regions, determine the pattern of brain damage [3,4,5], highlighting white matter injury, a main cause of periventricular leukomalacia (PVL), a leukodystrophy associated with PA [6].

PA involves two events, hypoxia and re-oxygenation. The primary energy failure leads to a metabolic crisis and, if sustained, cell death [7,8,9]. Re-oxygenation, a requirement for survival, implies glutamate-dependent excitotoxicity, mitochondrial dysfunction, accumulation of free radicals, and neuroinflammation [10,11,12], triggering a secondary energy failure and delayed cell death [8,9]. Delayed cell death is the most important endpoint for any therapeutic strategy [4,5,8,9,13], providing a window of opportunity for neuroprotection [14].

Although the pathophysiology of neuronal death following PA has been extensively studied, there is less attention paid to the role of glial cells, specifically oligodendrocytes (OLs) [15]. In the central nervous system (CNS), mature OLs are the cells that form myelin, which consists of oligodendroglial plasma membrane loops tightly woven concentrically around the axons [16], allowing the fast saltatory conduction of an action potential. OLs play a role in growth, stability, and axonal maintenance [17,18], and participate in the establishment and consolidation of neuronal networks, suggesting important roles during brain development, before myelination is established [19]. Thus, oligodendroglial injury and myelination deficiency may disrupt CNS development, interfering with axonal function, neuronal survival, and neurocircuitry consolidation, leading to motor, sensory, and cognitive deficits, depending upon the affected systems [20].

The endpoint of the differentiation program for the oligodendroglial lineage is the formation of myelin, which implies a transit from oligodendrocyte precursor cells (OPCs) to mature myelinating OLs. Each step in the oligodendroglial lineage is precisely regulated by intrinsic and extrinsic factors, ensuring correct myelination at the right location and right time [21], and allows for proper formation of myelin sheaths, consisting of lipids, in particular cholesterol, and galactolipids, and several proteins, including proteolipid and myelin basic protein (MBP) [22,23,24]. MBP, one of the main protein components of myelin, constitutes over 30% of the total protein content of the CNS [24]. MBP is the only structural protein found, so far, to be essential for myelin formation, presumably because of its role for myelin membrane compaction [25], thus maintaining the correct structure of myelin, and interacting with the lipids of the membrane [26]. Although myelination is primarily driven by different lineages of OLs, other glial cells contribute to myelination of the developing brain under homeostatic conditions, and to myelin repair [27,28,29,30,31,32]. Many lines of evidence support the idea that microglial activation and astrocytic reactions impair survival and maturation of OPCs, also leading to hypomyelination [33].

Olig-1 and Olig-2 are transcription factors involved in the modulation of OLs function under physiological and pathological conditions [34]. Olig-1 is involved in the repair of demyelinating lesions, stimulating OLs maturation, while Olig-2 directs the process toward OLs differentiation [35]. There are reports indicating that neuroinflammation increases the proliferation of OPCs, producing long-lasting defects in oligodendroglial maturation and myelination, likely by disruption of OL transcription factors [36,37]. In addition, proinflammatory cytokines released from activated microglia and/or astrocytes following brain injury are detrimental to the survival of OLs [38]. The balance of pro- and anti-inflammatory cytokines is likely affecting OPC differentiation. Indeed, Bain et al. showed that altered cytokine signaling contributes to the imbalance observed in the production of glial cells from the neonatal subventricular zone (SVZ) by hypoxia-ischemia (HI), promoting differentiation of glial progenitors towards astrocytes at the expense of developmentally more appropriate OLs [39].

OLs and their precursors are highly susceptible to metabolic insults, because of their neurochemical features (e.g., low antioxidant glutathione, high intracellular iron stores, high production of hydrogen peroxide, expression of AMPA receptors lacking GluR2 subunits), differentiation programs, as well as high metabolic rate and ATP requirements for the synthesis of large amounts of myelin [40,41,42]. Renewal of OLs and myelination continues throughout adult life [36,37], probably as a plastic mechanism to respond to brain injury [43,44,45].

There is evidence indicating that vulnerability of OLs depends upon their maturity. For example, pre-oligodendrocytes (pre-OL) are susceptible to metabolic insults occurring at birth [46,47], affecting cerebral myelination at later ages [48]. It has been reported that postnatal ischemia induces premature oligodendrogenesis and OPC migration to the site of damage, interfering, however, with OLs maturation [49], and decreasing myelin production [49,50,51]. In agreement, postnatal ischemia decreases the number of mature OLs, and myelination, impairing white matter integrity in the infarct area [49,52]. Kohlhauser et al. reported that PA has a long-term effect on myelination, observing a patchy myelin aggregation in the hippocampal fimbriae and cerebellum at adulthood. The mechanism by which PA affects myelination has not yet been established [53].

PA is an important pediatric issue with few therapeutic alternatives. Only hypothermia has shown relevant effects, but this treatment is limited by a narrow therapeutic window [54] and is still waiting for consensus on clinical protocols [6]. The use of mesenchymal stem cells (MSCs) has been proposed as a promising therapeutic strategy to manage complex diseases [55], like hypoxic-ischemic encephalopathy (HIE), specifically PVL [56]. MSCs are able to sense the microenvironment, changing their secretion patterns, according to the special requirements of the damaged tissue [55]. MSCs can release immunomodulatory and neurotrophic paracrine factors, promoting endogenous regeneration [57,58,59]. In postnatal ischemia models in rodents, it has been shown that MSCs treatment decreases the expression of pro-inflammatory molecules, such as IL-1β and IL-6 [60,61,62], and the number of activated astrocytes and microglial cells, suggesting immunomodulation [61,62,63,64]. van Velthoven et al. and Wei et al. showed that administration of hypoxia-preconditioned MSCs promotes angiogenesis, neurogenesis, and oligodendrogenesis, enhancing myelination and local cerebral blood flow in ischemic tissue [57,58,61].

Thus, we have investigated whether there is a sustained myelin deficit and OLs injury induced by PA up to P14, a critical postnatal period in the rat, characterized by a spur of networking [4,5,65,66]. The effect of neonatal MSCs treatment was evaluated, focusing on survival of mature OLs and myelination in telencephalic white matter.

## 2. Results

### 2.1. Apgar Scale

Table 1A,B shows the outcome of PA evaluated by an Apgar scale adapted for rats [67], applied 40 min after delivery up to P14. While 100% of control rat neonates (CS) initiated pulmonary breathing as soon as the amniotic tissue was removed from the animal nose, asphyxia-exposed neonates (AS) had to be cleaned and repetitively stimulated to respiration until initiating a first breath, which could or could not be sustained, supported by forced gasping until breathing was stabilized. The rate of survival shown by AS animals was approximately 60%, while it was 100% among CS. Surviving AS animals showed decreased respiratory frequency, decreased vocalization, blue (cyanotic) skin coloration, rigidity, and akinesia, indicating a severe insult. When received by surrogate dams, no significant differences were observed on survival rate among AS and CS rat neonates. However, there were some signs of a sustained physiological deficit, mainly affecting respiratory parameters. AS neonates showed a decreased respiratory frequency, observed at P1 (by ~20%), P7 (by ~20%), and P14 (by ~14%).

### 2.2. Effect of Neonatal Development and PA on Myelination in Telencephalon (External Capsule, Corpus Callosum, Cingulum) and Fimbriae of Hippocampus at P1, P7 and P14

Figure 1A–D shows representative microphotographs obtained by confocal microscopy showing DAPI (nuclei, blue) and MBP fluorescence (labelling OLs and myelin fibers, red) in external capsule (1A), corpus callosum (1B), cingulum (1C), and fimbriae of hippocampus (1D) of control (CS) and asphyxia-exposed (AS) rat neonates at P1, P7, and P14. A similar amount of DAPI positive nuclei was observed under all experimental conditions. No MBP labelling was seen at P1, but it was evident at P7 (as OLs or myelin fibers) and was significantly increased at P14 (mainly as compact myelin), both in CS and AS neonates. The progression of myelination was, however, different in the analyzed regions.

At P7, both OLs and myelin fibers were evident in external capsule, while at P14, OLs could no longer be seen as independent structures, because of the dense amount of MBP positive fibers, both in CS and AS animals. In the corpus callosum, OLs with long and branched processes were evident at P7, while at P14, as for external capsule, MBP positive fibers dominated over individual OLs. In the cingulum, some OLs and myelin fibers could be seen, but MBP fibers also dominated at P14. In the fimbriae of hippocampus, few OLs could be seen at P7, but again MBP fibers dominated at P14.

Table 2A,B shows the effect of PA on the number of DAPI (cells/mm^3^) and MBP positive pixels/total pixels quantified simultaneously in white matter of telencephalon (external capsule, corpus callosum, cingulum, and fimbriae of hippocampus), in AS and CS rat neonates at P1, P7, and P14. No significant differences in DAPI positive cells/mm^3^ among the different experimental conditions were observed, with around 30–40 × 10^4^ cells/mm^3^ for each condition at each age. No MBP immunostaining could be quantified in any of the studied regions at P1 sampled from CS and AS-exposed rats, but MBP immunostaining could be seen and quantified at P7. MBP-positive pixels were, however, decreased in the external capsule (by ~50%), corpus callosum (by ~60%) and cingulum (by ~70%) in AS compared to CS animals; no differences were observed in the fimbriae of hippocampus. In CS animals at P14 compared to P7, MBP immunostaining was increased in the external capsule (>2×), corpus callosum (>10×), cingulum (>6×), and fimbriae of hippocampus (>25×), reflecting a spur of myelination [68]. In AS, that spur was even more remarkable, increased by >5× (external capsule >5×; corpus callosum >30×; cingulum >20×, and fimbriae of hippocampus >30×).

### 2.3. Effect of Neonatal Development and PA on mRNA Levels (MBP, OL Transcription Factors, and Pro-Inflammatory Cytokines) Evaluated in the Telencephalon by RT-qPCR at P1, P7 and P14

Gene transcripts were amplified by RT-qPCR, quantified by the ΔΔCT method, using β-actin as housekeeping gene, normalizing the mRNA levels of each target genes to that observed in CS neonates at P1. Table 3A,B shows fold changes in mRNA levels, monitored in telencephalon of CS (3A) and AS (3B) rat neonates at P1, P7, and P14.

#### 2.3.1. MBP, Olig-1 and Olig-2 mRNA Levels

As already shown by immunocytochemistry, the MBP mRNA levels increased along development in CS neonates, by 20× at P7, and by >1000× at P14 (Table 3A). A similar increase was observed in AS neonates (Table 3B). Olig-1 and Olig-2 mRNA levels increased in telencephalon at P14 in CS, but only Olig-1 mRNA levels increased in AS neonates along development. Olig-1 mRNA levels were already increased at P7 in AS (Table 3B), compared to that in CS (>1.4×) (Table 3A). 

#### 2.3.2. Pro-Inflammatory Cytokines (IL-1β, IL-6, TNF-α, Cox-2) mRNA Levels

In CS animals (Table 3A), IL-1β (>2×) and IL-6 (>1.8×) mRNA levels increased at P14, while no significant differences were observed in TNF-α mRNA levels. Cox-2 levels increased along development, both in CS and AS animals (P7, >5× and P14, 20×). In AS neonates, IL-6 and TNF-α mRNA levels increased along development, with a maximum observed at P14 (>1.8× and >18×, respectively), but no significant differences were observed in IL-1β mRNA levels (Table 3B). At P7 an increase in IL-6 mRNA levels was observed in AS animals, but no differences were observed in IL-6 protein levels (F (2, 23) = 0.194, *p* = 0.825; see Appendix A).

### 2.4. Effect of PA on Glial Cells Measured in Telencephalon and Fimbriae of Hippocampus at P7, Focusing on Mature OLs (MBP), Astrocytes (GFAP) and Microglia (Iba-1)

The effect of PA on mature OLs, astrocytes, and microglia was further investigated in an independent cohort of CS and AS animals analyzed at P7, the only time when individual OLs could be observed.

Figure 2A–D shows microphotographs of OL (MBP+), astroglia (GFAP+), and microglia (Iba-1+) in the external capsule (Figure 2A), corpus callosum (Figure 2B), cingulum (Figure 2C), and fimbriae of hippocampus (Figure 2D) of CS and AS at P7. In adjacent microphotographs, OLs (red), astrocytes (red), and microglia (green) are intermingled with DAPI (blue) labelling. Table 4A,B shows the quantification of DAPI+, MBP-DAPI+, GFAP-DAPI+, and Iba-1-DAPI+ cells/mm^3^. The number of DAPI+ nuclei at P7 was similar to that shown above in the different cohort of CS and AS neonates (cf. Table 2 versus Table 4). OLs, astrocytes, and microglia represented approximately 15% of the total number of DAPI+ cells observed in CS and AS. The number of MBP-DAPI+ cells/mm^3^ in AS at P7 was, however, decreased by approximately 50% compared to CS neonates. No significant differences were observed in GFAP-DAPI+ cells/mm^3^ or Iba-1-DAPI cells/mm^3^ in CS and AS neonates, apart from that in cingulum, where the number of Iba-1-DAPI+ cells/mm^3^ increased >1.5× in AS, compared to CS animals.

### 2.5. Effect of MSCs Treatment on Cell Death, Number of OL and Myelination at P7

Vehicle or MSCs treatment was administered i.c.v. two hours after delivery. Brain samples were taken at P7, focusing on external capsule, corpus callosum, and cingulum. The fimbriae of hippocampus were not evaluated since no differences in myelination among AS and CS were observed in that region (cf. Table 2A,B).

#### 2.5.1. Effect of PA on Apoptotic-Like (TUNEL-DAPI+/mm^3^) and OL-Specific Cell Death (TUNEL-DAPI-MBP+/mm^3^) at P7

Figure 3A–C shows microphotographs for the co-localization of TUNEL and DAPI in the external capsule (Figure 3A), corpus callosum (Figure 3B), and cingulum (Figure 3C) of CS and AS neonates at P7, treated with vehicle or MSCs.

As shown in Table 5A, the TUNEL-DAPI+/mm^3^ co-localization was estimated to be approximately 2% of DAPI+/mm^3^ in vehicle-treated CS neonates, and that proportion was duplicated in vehicle-treated AS neonates (Table 5C), indicating an increased cell death following PA. In vehicle-treated CS neonates, OL-specific cell death (TUNEL-MBP-DAPI+/mm^3^) was 1.9% in the external capsule, and 16% in the cingulum. No OL-specific cell death was found in the corpus callosum of CS animals (Table 5A). TUNEL- MBP-DAPI+/mm^3^ observed in external capsule, corpus callosum and cingulum of vehicle treated AS neonates was 4.5%, 2.8%, and 5.5% of the total TUNEL-DAPI/mm^3^, in external capsule, corpus callosum and cingulum, respectively, suggesting PA increased OL-specific cell death (Table 5C). In the cingulum however, the rate of OL-specific cell death was similar to that of vehicle-treated CS neonates (cf. Table 5C versus Table 5A).

#### 2.5.2. Effect of MSCs Treatment on Cell Death, Number of OLs and Myelination at P7

No differences were observed in the number of DAPI (TUNEL-DAPI+) or OL-specific (TUNEL-MBP-DAPI+) cell death, in vehicle- compared to MSCs-treated CS neonates, except in cingulum, where no oligodendroglial death could be detected in the MSCs-treated CS group (cf. Table 5A versus Table 5B). In the corpus callosum, no TUNEL-MBP-DAPI+ cells could be observed neither in vehicle- nor MSCs-treated CS neonates. It was evident, however, that MSCs treatment prevented the increase of TUNEL-DAPI+/mm^3^ and TUNEL-MBP-DAPI+/mm^3^ cell death observed in vehicle-treated AS neonates at P7 (cf. Table 5C versus Table 5D).

#### 2.5.3. Effect of PA and MSCs Treatment on the Number of Mature OLs (MBP-DAPI+/mm^3^) and Myelination (MBP-Positive Pixels/Total Pixels)

No differences were observed in the number of OLs (MBP-DAPI+/mm^3^) and myelination (MBP positive pixels/total pixels) in vehicle-compared to MSCs-treated CS neonates (cf. Table 6A versus Table 6B). When compared to CS, MBP-DAPI/mm^3^ and MBP pixels/total pixels were decreased in external capsule and cingulum of AS animals (cf. Table 6C versus Table 6A), but no differences were observed in corpus callosum among any of the groups (vehicle- or MSCs-treated, CS, or AS animals).

MSC treatment prevented the decrease observed in OL number (MBP-DAPI+/mm^3^) and myelination (MBP positive pixels/total pixels) in AS rat neonates, compared to vehicle AS neonates (cf. Table 6C versus Table 6D). Indeed, the number of MBP-DAPI+ cells was almost on par with vehicle- and MSC-treated CS animals (cf. Table 6D versus Table 6A).

Figure 4 shows the number of OL and myelination at P7. 

## 3. Discussion

This study is on the effect of perinatal asphyxia (PA) in rats, focusing on myelination and glial cell response, and on the potential of MSC treatment to ameliorate these effects. The effect of global PA on myelination and OL maturation is a highly clinically relevant issue. Common experimental models in rodents investigate the effect of a HI insult at postnatal stages [69], when the OPCs have already differentiated into mature myelinating OLs, likely resulting in a different outcome to when PA occurs at an earlier developmental stage, as explored in the present paper.

The effect of PA on myelination, expression of OL transcription factors, neuroinflammation, glial cells, and cell death was analyzed in the telencephalon of rats at P1, P7, and P14, evaluating the effect of neonatal MSC treatment to prevent damage. The effect of PA was monitored at a critical postnatal period characterized by a spur of neuronal networking, finding that the most relevant changes occurred at P7.

It was found that PA produced: (i) A decrease in MBP-immunoreactivity at P7 in the external capsule, corpus callosum, and cingulum, but not in the fimbriae of the hippocampus; (ii) an increase in Olig-1 mRNA levels at P7; (iii) an increase in IL-6 mRNA, but not in protein levels at P7; (iv) an increase in cell death, including OL-specific cell death at P7; and (v) MSCs treatment prevented the effect of PA on myelination, OL number, and cell death.

No MBP-labelling was observed in any of the analyzed regions at P1, under any of the experimental conditions. In CS and AS animals, OLs and/or myelin fibers were evident at P7, while at P14, the soma of OLs could not be seen as independent structures, because of the dense amount of MBP positive fibers. At P7 and P14, MBP immunoreactivity increased, both in CS and AS neonates, mainly in the external capsule, cingulum, and corpus callosum, regions myelinated at earlier stages than the fimbriae of hippocampus, also investigated in the present report. This agrees with the idea that myelination progresses in an age- and region-dependent manner [51]. Indeed, myelination occurs in response to functional demands, such as suckling reflex, and other motor and sensory requirements related to early nursing and behavioral skills [68]. It was found that PA induced a decrease in myelination and OLs number at P7, likely related to the long-term white matter dysfunction reported at adulthood by Kohlhauser et al. [53]. During development, OLs also play an important role in the formation and sculpturing of neurocircuits, exerting regulatory functions, guiding, and stabilizing neuronal connections, regardless of the onset of myelination, and preventing aberrant connections at adulthood [70,71]. It was previously reported that the neonatal genetic ablation of OLs during the first postnatal week induced a severe structural and functional impairment of the cerebellar cortex, even after recovery of OLs and myelination [19]. Thus, the long-term changes in white matter observed following PA can be explained by interruption of an early OLs regulatory function.

The endpoint of the oligodendroglial lineage is the formation of myelin, involving the differentiation and maturation of neural stem cells from the SVZ to myelinating OLs [72]. Interestingly, the maximal vulnerability to PVL occurs at a period before the onset of active myelination [46], which in humans is around post-conceptional week 28–32 [73]; equivalent to P2-5 in rats [74] when myelination starts, performed by mature OLs. In rats, myelination peaks by the second and third postnatal weeks, and continues into adulthood, albeit at a lower rate [19]. Therefore, a perinatal metabolic insult would result in hypomyelination, while a postnatal insult would lead to white matter damage [75], or failure of OPCs to generate a new pre-OL pool, in response to neuroaxonal degeneration [76]. Thus, in the present study, we focused on a developmental window equivalent to a preterm human baby, when the pre-OL phenotype predominates, and thus the highest risk period for the establishment of the white matter [74].

Prematurity is a relevant issue. There are several studies reporting that vulnerability to HI depends on the maturity of the oligodendroglial lineage, with pre-OL being the most vulnerable phenotypes, [46,47]. In the rat at P7, immature OLs predominate [76] and are more resistant to cell death induced by HI hypoxia than pre-OL, but similar to that of mature OL [47]. However, a metabolic insult at a delayed developing phase can still interrupt myelination, by acting on areas showing persistence of the pre-OL phenotype, as observed in cerebral cortex [76].

Clinical and experimental studies [77] have shown that the long-term consequences of PA depend on the severity and duration of the insult, as well as on the developmental stage of the affected region, implying differences in antioxidant defenses, local metabolic imbalance during the re-oxygenation period, and rescue mechanisms [78]. Thus, the ability of OPC proliferation and differentiation to overcome metabolic deficits might explain the regional differences observed in myelination. We previously reported in vivo [79] and in vitro [80,81,82] experiments that PA increases BrdU-positive cell proliferation in hippocampus and SVZ, but only approximately 40% of BrdU-positive cells were double-labelled with the neuronal marker MAP-2, suggesting a predominance of gliogenesis. However, gliogenesis does not necessarily imply oligodendrogenesis [39]. A newly generated population of OL may explain why no differences in myelination were observed between CS and AS animals in any of the analyzed regions at P14. The possibility also exists that an excess of OPC is generated during ontogenesis, remaining at an undifferentiated stage, forming part of the CNS parenchyma along the lifespan, mobilized in response to pathological signals, contributing to restoration of axon myelination [51]. Therefore, more studies are required to evaluate the effects of global PA on OL precursors during neurodevelopment.

Regulation of the oligodendroglial lineage is a multifactorial process. Basic helix–loop–helix transcription factors Olig-1 and Olig-2 play an essential role in oligodendroglial development [34]. These transcription factors are coordinately expressed along development [83], playing a pivotal role in biological processes, including OL proliferation, differentiation, maturation, and myelination/remyelination [84,85]. Persistent expression of Olig-2 throughout oligodendroglial development has been reported, suggesting that this factor is involved in OPC specification and differentiation, repressing OL maturation and myelination [85,86]. Olig-1 factor is an important marker for OPC [87,88], and it can stimulate differentiation and maturation to OLs [83], as well as OPC repair. Nevertheless, in the present study we found that Olig-1, but not Olig-2 mRNA levels increased in the telencephalon of AS neonates compared to CS neonates at P7. However, controversial results have also been reported. French et al. showed that oxidative stress following postnatal HI reduced the expression of Olig-1 [89], while Cheng et al. reported a decrease in Olig-1 protein levels 1-3 days after postnatal HI, but an increase when measured 7-21 days after HI, although levels failed to reach those observed under normoxic conditions [90].

Pro-inflammatory cytokines (IL-1β, IL-6, TNF-α, Cox-2) have been reported to produce cytotoxic effects on OL, in vitro [91,92,93] and in vivo [37,49], inducing premature maturation of the existing precursors and aberrant myelination, thereby contributing to hypomyelination. In this study, it was observed that IL-6 mRNA levels increased only 1.6 times in AS with respect to CS neonates at P7, but not in protein levels. We suggest, therefore, that IL-6 mRNA expression using RT-qPCR to total RNA from telencephalic brain samples is more sensitive than directly measuring protein levels. Similar effect was observed by Neira-Pena et al., who showed that PA increased several pro-inflammatory cytokines, but not necessarily increased protein levels [94,95]. Changes in pro-inflammatory cytokines were observed in the hippocampus 8 h after PA [94], but in mesencephalon, up to 24 h after the insult [95], which could be explained by temporally and regionally selective mechanisms.

Activation of microglia and astrocytes is one of the signatures of neuroinflammation [96]. Hence astrocytes and microglial cells were also assessed, since glia–glia crosstalk plays an important role in the modulation of OL homeostasis, along with myelination, demyelination, and re-myelination [97]. Astrocyte activation is an early indicator of neuropathology, measured as increased GFAP expression and associated linked morphological alterations [98,99]. No changes were observed in the number of GFAP-DAPI+ cells/mm^3^ in the external capsule, corpus callosum, cingulum, and fimbriae of hippocampus, in agreement with previous reports [15]. Nevertheless, in the present study, astrocyte morphology was not assessed, therefore a more detailed study is required to evaluate glial cell activation induced by global PA. Microglial cells are the primary effectors of the innate immune response within the CNS, and their activation occurs at early stages of disease, often preceding overt neuropathology [100,101]. In the present study, no changes were observed in Iba-1-DAPI+ cells/mm^3^ in the external capsule, corpus callosum, or fimbriae of the hippocampus, but an increase was observed in the cingulum at P7. These results support the proposal that OLs are the most vulnerable glial cells to hypoxia-reoxygenation insults, at least in the examined brain regions. Nevertheless, the number of microglial cells is not necessarily indicative of neuroinflammation, but it is the activation levels of microglia, which can be assessed, either with markers such as CD11b, or using a ramification index [102].

TUNEL/DAPI co-labelling indicates cells death, mainly at the DNA fragmentation stage, considered to be an early event of apoptosis [103], although postnatal HI triggers an apoptosis-necrosis continuum [8]. In the present study, it was found that there was a decrease in the number of OL and an increase in total and OL-specific cell death in telencephalic regions, suggesting a developmental disruption in differentiation processes, but also direct damage to the mature OL phenotype. The multistep process of oligodendroglial lineage differentiation is highly energy-dependent and regulated by the expression of numerous genes susceptible to hypoxia [56]. Indeed, OLs are vulnerable to metabolic insults, oxidative stress, inflammation, and/or excitotoxicity [40,42].

Several lines of evidence indicate that pre- and immature OL undergo a reactive response to neonatal HI, characterized by morphological changes, involving cell soma and processes, with a distribution that overlaps that of reactive astrocytes and reactive microglia [47]. It has been shown that there is an accelerated maturation of damaged pre-OL to immature OLs [50,104,105,106], which may be maladaptive, delaying or even arresting differentiation and maturation [49,50,80], causing hypomyelination. Therefore, it would be relevant to evaluate in future studies the effects of global PA on the morphology of OLs and their precursors.

There is extensive preclinical evidence indicating that MSCs administration can be an effective option to prevent neurodegenerative cascades, by altering OL differentiation and/or preventing cell death [56]. In the present study, we found that MSC treatment significantly increased MBP levels in the external capsule and cingulum.

It is suggested that MSC treatment decreases brain damage by either (i) inhibiting injurious signals; (ii) replacing lost tissue; and/or (iii) enhancing endogenous repair processes [107,108]. Thus, several reparative mechanisms can take place in a region- and time-dependent manner, expecting therefore differential effects by MSC treatment.

The present results agree with those reported by van Velthoven et al., who demonstrated that a single, acute administration of MSCs in neonatal HI rats restored OL number and myelination in telencephalon [57,58]. Furthermore, inhibition of deleterious processes such as neuroinflammation may also contribute to the beneficial effects of MSC treatment in restoring myelination [107]. Although we did not evaluate the mechanisms by which MSC treatment prevents OL loss and deficits in myelination, there is evidence that MSCs can modify their secretome in situ, releasing neurotrophic factors in a paracrine fashion, thereby promoting endogenous repair and regeneration [57,58,108]. Interestingly, a recent study published by our laboratory evaluated the effects of intranasal administration of mesenchymal stem cell-derived secretome (MSC-S) [109]. Unlike the use of MSCs as a therapeutic strategy, MSC-S can show better safety, dosage, and potency similarly to conventional pharmaceutical drugs, be readily available since it can be lyophilized, favoring storage, with low factoring costs for large-scale production reutilizing the yielding cells [110]. The reported study showed that MSC-S reversed oxidative stress induced by PA in the hippocampus, increasing antioxidative Nuclear Erythroid 2-Related Factor 2 (NRF2) translocation, and NQO1, an antioxidant protein [109]. Furthermore, MSC-S reduced neuroinflammation by decreasing nuclear NF-κB/p65 levels and microglial reactivity, in association with decreased cleaved-caspase-3 cell-death [109]. MSCs can also induce several anti-apoptotic mechanisms, up-regulating DNA repair, and down-regulating mitochondrial death pathways, increasing antioxidant activity and altering the expression of anti- and pro-apoptotic proteins [111]. Thus, whether a single administration of MSCs protects against the loss of mature OLs and delayed myelination elicited by PA in telencephalic white matter remains to be investigated, also whether MSCs act via an antioxidant or anti-inflammatory mechanism, involving NRF-2 activation and other molecular pathways.

## 4. Materials and Methods

### 4.1. Animals

Wistar rats from the animal station of the Molecular & Clinical Pharmacology Programme, ICBM, Faculty of Medicine, University of Chile, Santiago, Chile, were used for all experiments. The animals were kept at a temperature- and humidity-controlled environment with a 12/12 h light/dark cycle, with access to water and food ad libitum when not used in the experiments, and the well-being of the animals was monitoring monitored continuously by qualified personnel.

### 4.2. Ethic Statement

All procedures were conducted in accordance with the animal care and use protocols established by a Local Ethics Committee for experimentation with laboratory animals at the Medical Faculty, University of Chile (Protocol CBA#0943 FMUCH), approved by the Medical Faculty, University of Chile, and by an ad-hoc commission of the Chilean Council for Science and Technology Research (CONICYT) (FONDECYT #1120079, 2012; #1190562, 2019), endorsing the principles of laboratory animal care (NIH; N° 86-23; revised 1985). Animal wellbeing was continuously monitored (on 24 h basis), following the ARRIVE guidelines for reporting animal studies (www.nc3rs.org.uk/ARRIVE (accessed on 20 March 2021)).

### 4.3. Perinatal Asphyxia

Pregnant Wistar rats within the last day of gestation (G22) were euthanized and hysterectomized. Two or three pups per dam were removed immediately and used as non-asphyxiated caesarean-delivered controls (CS). The remaining fetuses within the uterine horns were immersed into a water bath at 37 °C for 21 min (asphyxia-exposed rats, AS). Following asphyxia, the uterine horns were incised, and the pups removed, stimulated to breathe, and after an approximately 40 min observation period on a warming pad, were evaluated with an Apgar scale for rats, according to Dell’Anna et al. [67], and then nurtured by a surrogate dam. The physiological parameters (e.g., respiratory frequency, vocalization, skin coloration, movements, and coordination) were also monitored up to P14, comparing the same CS and AS cohorts.

### 4.4. Isolation, Expansion, and Characterization of Rat Adipose Tissue-Derived MSCs

Wistar 12-week-old female (220–250 g) were anesthetized and euthanized. For isolation of MSCs, dorsal subcutaneous fat was dissected, washed with phosphate-buffered saline (PBS), and cut into small pieces. Tissue was digested with 1 mg/mL collagenase type II (Gibco, Grand Island, NY, USA) in PBS, incubated under agitation at 37 °C for 90 min. At the end of digestion, 10% fetal bovine serum (FBS; Gibco, Auckland, New Zealand) was added to neutralize collagenases. The mixture was centrifuged at 400× *g* for 10 min to remove floating adipocytes. Pellets were resuspended in alpha-minimum essential medium (α-MEM; GIBCO), supplemented with 10% FBS and 0.16 mg/mL gentamicin (referred to as α-10 medium), plated at a density of 7000 cells/cm^2^. Cells were cultured at 37 °C in a 5% CO_2_ atmosphere. When foci reached 80% confluence, cells were detached with 0.25% trypsin (Sigma-Aldrich, St. Louis, MO, USA), centrifuged and subcultured at 7 × 10^3^ cells/cm^2^. The cells were frozen at −80 °C in a cryopreservation medium, pending further experiments.

After 2 subcultures, adherent cells were characterized according to adipogenic, osteogenic, and chondrogenic potential, as previously described [55,112] (Appendix A). The cells were incubated with standard adipogenic (1 μM dexamethasone and 10 μM rosiglitazone for 14 days), osteogenic (0.1 μM dexamethasone, 50 μg/mL L-Ascorbate 2-phosphate, and 10 mM b-glycerol phosphate for 21 days), and chondrogenic (0.1 μM dexamethasone, 0.1 μg/mL L-Ascorbate 2-phosphate, 0.5 UI/mL insulin, and 10 ng/mL transforming growth factor beta-3 for 10 days) differentiation media. To evaluate the adipogenic potential, cultures were stained with 60% Oil Red O for 1 h (Sigma-Aldrich). To evaluate the osteogenic potential, cultures were fixed with 70% ethanol for 30 min and stained with 40 mM Alizarin Red (Sigma-Aldrich) for 10 min. To evaluate the chondrogenic potential, cultures were fixed with 70% ethanol for 10 min and stained with 0.1% Safranin O (Sigma-Aldrich) for 5 min. Once washed, cells were observed and photographed by a light microscope (Eclipse TS100, Nikon, Japan). Immunophenotyping was performed by flow cytometric analysis after immunostaining with monoclonal antibodies against the putative murine MSCs markers CD-29 (FITC-conjugated) and CD-90 (PE-conjugated), or characterized for markers of hematopoietic cell lineages, CD-45 (APC-conjugated) and CD-11b (APC-conjugated). All antibodies were purchased from BD Biosciences (San Diego, CA, USA).

### 4.5. Administration of MSCs

Two hours after delivery, AS and CS neonates were randomly assigned to receive: (i) A single intracerebroventricular (i.c.v.) administration of vehicle (5 µL of 10% rat plasma in saline [113] (CS or AS vehicle, n = at least 5), or (ii) a single i.c.v. administration of MSCs (5 µL of 5 × 10^4^ cells; CS or MSCs-treatment, n = at least 5). The i.c.v. route of administration was chosen to attain a proof of principle, regarding an effect occurring within the CNS. The i.c.v. treatment was performed under cryoanaesthesia with a cannula implanted 1 mm lateral to Bregma, 2 mm deep under the scalp, injecting slowly with a 50-μL Hamilton syringe and a properly prepared injection cannula (0.6 mm of diameter, with a sharpened sharped tip less than 0.2 mm of diameter), connected with a dialysis tubing, kept manually in place for 2 min after the injection was performed. The neonates were continuously observed until given to a surrogate dam and euthanized at P7, focusing on external capsule, corpus callosum, and cingulum.

### 4.6. Tissue Sampling for Immunofluorescence

Sampling and preparation of brain tissue for immunohistochemistry was performed according to Morales et al. [82]. Briefly, CS and AS rats at P1, P7, or P14 (basal, vehicle or MSCs-treatment conditions; N = at least 5) were anesthetized and perfused transcardially with 0.1 M PBS (pH 7.4), followed by 4% paraformaldehyde in 0.1 M PBS (pH 7.4). The brain was removed from the skull, post-fixed in the same fixative solution overnight at 4 °C and immersed in 10% sucrose in 0.1 M PBS for 2 days and subsequently in 30% sucrose at 4 °C for 2 days. Coronal sections (20 μm thick) of the telencephalon (between Bregma 1.20 and 0.60 mm at P1; at 0.60 and 0.20 mm at P7; −0.40 mm and −0.20 mm at P14) and hippocampi (between Bregma −0.40 and −1.00 mm at P1; −1.40 and −2.00 mm at P7; −1.60 and −2.40 mm at P14; https://www.ial-developmental-neurobiology.com/en/publications/collection-of-atlases-of-the-rat-brain-in-stereotaxic-coordinates, accessed on 20 March 2021)) were obtained using a cryostat (Thermo-Fischer Scientific Microm HM 525, Waldorf, Germany) and processed for immunofluorescence (IF) to detect myelinated fibers and mature OLs (MBP), astrocytes (GFAP), and microglia (Iba-1).

### 4.7. Antibodies

#### 4.7.1. Primary Antibodies

(i) Anti-MBP (myelinated fiber and mature OL marker, chicken, Aves Labs Inc., Tigard, OR, USA, #MBP; 1:750 in blocking solution containing 1% bovine serum albumin (BSA), 5% normal goat serum (NGS) and 0.3% Triton X 100). (ii) Anti-GFAP (astrocyte marker, mouse, Sigma-Aldrich, Life Science, Darmstadt, Germany, #3893; 1:500 in blocking solution containing 10% NGS and 0.3% Triton X 100); and (iii) anti-Iba-1 (microglial cell marker, rabbit, WAKO, Malaysia, #019-19741; 1:500 in blocking solutions containing 1% BSA, 10% NGS and 0.3% Triton X 100).

#### 4.7.2. Secondary Antibodies

(i) Anti-chicken-Alexa594 (goat, Invitrogen, Thermo Fisher, Scientific, #A-11042; 1:400 incubated in blocking solution containing 1% NGS); (ii) anti-mouse-Alexa488 (goat, Invitrogen, #A-11001; 1:500 incubated in blocking solution containing 0.3% Triton X100; and (iii) anti-rabbit-Alexa488 (goat, Invitrogen, #A-11034; 1:500 incubated in blocking solution containing 1% NGS).

### 4.8. Immunofluorescence

Coronal sections were rinsed with 0.1 M PBS and treated with blocking solutions for 1 h, incubated with primary antibodies (MBP, GFAP, or Iba-1) in blocking solution overnight at 4 °C in darkness. After repeated rinsing with 0.1 M PBS, samples were incubated with the corresponding secondary antibodies and counterstained with 4,6 diamino-2-phenylindol (DAPI, Invitrogen, 0.02 M; 0.0125 mg/mL; for nuclear labelling) for 2 h. After rinsing, the samples were mounted with Fluoromount and examined by confocal microscopy (Olympus-fv10i, Center Valley, PA, USA).

### 4.9. TUNEL Assay

Preparation of brain tissue for TUNEL assay was performed according to Perez-Lobos et al. [15]. Coronal sections fixed with 4% paraformaldehyde were stained with the ApoTag Peroxidase in Situ Apoptosis Detection Kit (MILLIPORE, Sigma-Aldrich, # 90418) according to manufacturer’s instructions. Coronal sections were rinsed three time with 0.1 M PBS, permeabilized with a solution containing 50% ethanol and 50% acetic acid for 5 min at 20 °C and treated with 3% H_2_O_2_ in methanol for 10 min for endogenous peroxidase quenching. Sections were then washed with a commercial solution provided by the manufacturer of the TUNEL assay, and incubated with TdT Enzyme at 37 °C for 80 min. The reaction was stopped with Stop Wash Buffer and incubated with a Ready-to-Use-anti-digoxigenin HRP-conjugated antibody for 35 min. After rinsing, the samples were incubated with 488-conjugated tyramide (1:100 diluted in 0.1 M PBS), as substrate for horseradish peroxidase (HRP) enzyme (Sigma-Aldrich), at room temperature for 10 min and then rinsed with 0.1 M PBS.

### 4.10. Image Processing and Stereological Analysis

The following parameters were assessed: (i) Myelination, for which a fixed intensity threshold for each area was defined, calculating the ratio of positive stained pixels for MBP, over the total pixel number, as previously described [114]; (ii) number of MBP positive cells/mm^3^ as previously described [15,79], counterstained for DAPI (nuclear staining) in all cases; (iii) number of GFAP-DAPI positive cells/mm^3^; (iv) number of Iba-1-DAPI positive cells/mm^3^; (v) number of TUNEL positive cells/mm^3^, for quantification of apoptotic-like DNA fragmentation; and (vi) number of TUNEL-MBP-DAPI positive cells/mm^3^.

Microphotographs were taken from highly myelinated areas of telencephalon and hippocampus from both hemispheres, focusing on: (i) external capsule (10 fields); (ii) corpus callosum (5 fields); (iii) cingulum (2 fields); and (iv) fimbriae of hippocampus (4 fields), with an Olympus confocal FV10i microscope using 60× objective lens (NA1.30). The area inspected for each stack was 0.04 mm^2^. The thickness (*Z*-axis) was measured, and the volume of the tissue sample was determined for each case. The number of positive stained cells for each marker was then counted in the volume of tissue using ImageJ software, confirmed by proper DAPI labelling, and counts were expressed as cells/mm^3^ in samples from CS and AS (basal, vehicle or MSCs-treated conditions). MBP, GFAP, and Iba-1 positive cells were counted manually when immunoreactivity overlapped at three levels through a section (Z-step 1 µm). An investigator blinded to the treatment made the respective quantifications for all parameters.

### 4.11. RT-qPCR and Enzyme-Linked Assay (ELISA)

#### 4.11.1. Tissue Sampling

The animals (basal, vehicle or MSCs-treated conditions; N = at least 5) were euthanized at P1, P7, or P14 by decapitation. The brain was quickly removed, and the telencephalon and hippocampus dissected on ice, using a newborn rat brain slicer (Zivic Instruments Pittsburgh, Pittsburgh, PA, USA). The samples were stored at −80 °C until required.

#### 4.11.2. RT-qPCR

Homogenization of tissue, extraction, and quantification of RNA for RT-qPCR: Telencephalon samples at P1, 7, and 14 of CS and AS neonates under basal conditions were homogenized with a Cordless Pellet Pestle homogenizer (Kimble Chase, DWK, Life Sciences, Rockwood, TN, USA). Total RNA was purified using TRIzol reagent (Invitrogen, Thermo-Fisher). The concentration and purity of total RNA was determined for optical density 260/280 absorption ratios. RNA integrity was evaluated by denaturing gel electrophoresis. One microgram of total RNA was used to perform reverse transcription with MMLV reverse transcriptase (Invitrogen) and oligo dT primers. RT-qPCR reactions were performed to amplify: (i) Olig-1 and Olig-2; (ii) MBP, and (iii) the inflammatory cytokines IL-1β, IL-6, TNF-α and COX-2, using a Light-Cycler 1.5 thermocycler (Roche, Indianapolis, IN, USA). RT-qPCR was performed using 2X Brilliant III SYBR Green QPCR Master Mix (Agilent Technologies, Santa Clara, CA, USA) in a MX3000 system (Stratagene, La Jolla, CA, USA). Primer sequences used for amplification are indicated in Appendix A. β-Actin was chosen as the housekeeping gene, based on the similarity of mRNA expression across all sample templates. Relative quantification was performed by the ΔΔCT method.

#### 4.11.3. ELISA

Homogenization of tissue and protein quantification for ELISA: Telencephalon obtained at P7 from CS and AS neonates, under vehicle and MSCs-treated conditions was lysed and homogenized with a Cordless Pellet Pestle homogenizer (Kimble Chase) in RIPA lysis buffer (Thermo, Waltham, MA, USA), supplemented with protease inhibitors (Thermo, Waltham, MA, USA). The lysates were incubated under agitation at 4 °C for 20 min, centrifuged at 8000× *g*, 4 °C, for 5 min. The supernatant was transferred to fresh Eppendorf tubes, and stored at −80 °C, pending further experiments.

Total protein concentration was determined by a commercially available Microplate bicinchoninic acid (BCA) protein assay kit-reducing agent compatible from Pierce (Thermo Fisher Scientific). Absorbance was measured at 562 nm in a Multi-Mode Microplate Reader (Synergy HT Biotek Instruments, Inc., Winooski, VT, USA).

Quantification of IL-6 protein levels was performed by a Rat IL-6 ELISA kit (BMS625; Invitrogen) following the manufacturer’s instructions. Briefly, 50 μL of standard, control, or sample was added into microplates coated with a monoclonal antibody against IL-6 and incubated with a biotin-conjugated anti-rat IL-6 antibody for 2 h at room temperature in a shaker at 400 rpm. Plates were washed and incubated with 100 μL of Streptavidin conjugated with HRP for 1 h at room temperature in a shaker at 400 rpm. The plates were rinsed and incubated with 100 μL of the substrate solution, containing hydrogen peroxide (H_2_O_2_) and tetramethylbenzidine as chromogen for 30 min at room temperature. The reaction was stopped, and optical density was determined using a Multi-Mode Microplate Reader (Biotek Instruments, Inc., Winooski, VT, USA) set at 450 nm. The concentration of protein levels was calculated by interpolation from a standard curve for IL-6 (from 8 standard dilutions; 31.3–4000 pg/mL), using a five-parameter curve fit model (MyAssay software).

### 4.12. Statistics

Data analysis was performed with a XLSTAT software, version 2019. 16.0.11601 (Addinsoft Sarl, Paris, France), allowing unbalanced multiple comparisons. Whenever the data suggested a normal distribution and/or similar variance (δ2), parametric comparisons were performed by *t*-test or two-way ANOVA followed by Benjamini–Hochberg correction as a post hoc test for multiple comparisons (in parentheses, the degree of freedom [d.f.] for each comparison, revealing the exact number of observations used for each case). All results are expressed as means ± standard error of the means (SEM). For statistically significant differences, the probability of error was set up to less than 5%.

## 5. Conclusions

The present findings provide evidence that PA induces regional and developmental-dependent changes on myelination and OL maturation. OLs appear to be the most vulnerable glial cells to hypoxia-reoxygenation insults at early neonatal stages, evaluated in vulnerable areas. Early neonatal MSCs treatment improves survival of mature OLs and myelination in telencephalic white matter. The outcome occurs in association with decreased cell death.

## Figures and Tables

**Figure 1 ijms-22-03275-f001:**
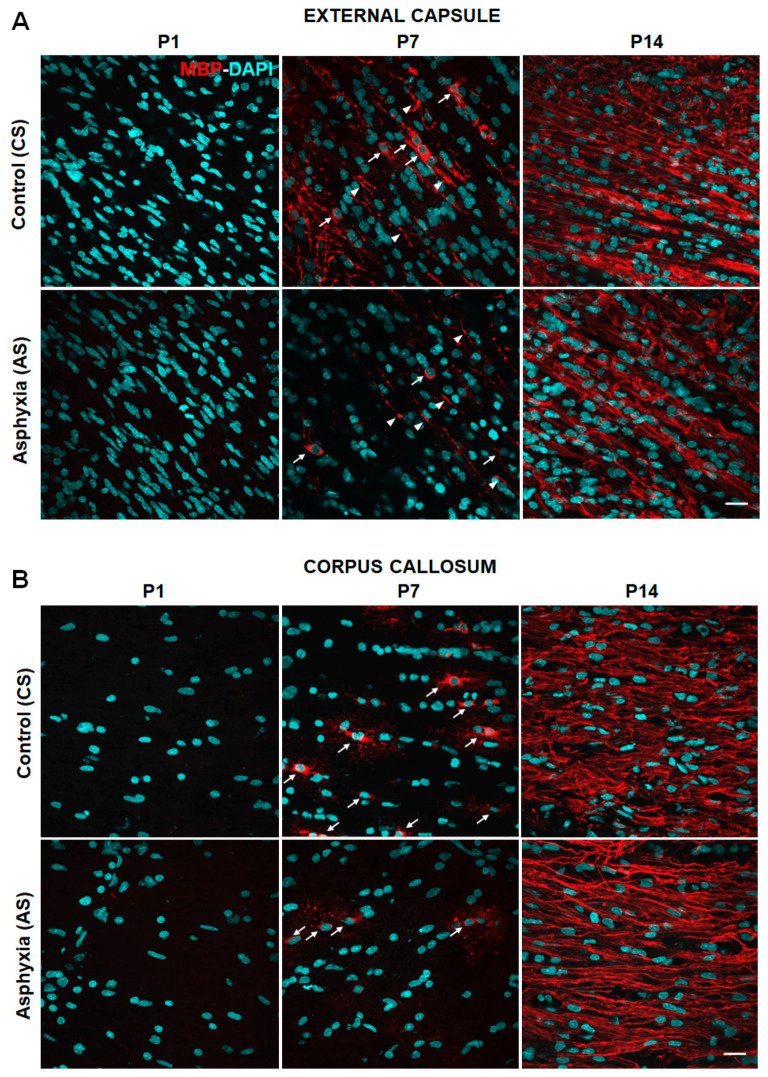
Effect of perinatal asphyxia (PA) and neonatal development on myelination at P1, P7, and P14, measured in external capsule (**A**); corpus callosum (**B**); cingulum (**C**) and fimbriae of hippocampus (**D**) of rat neonates. Representative microphotographs obtained by confocal microscopy showing myelin basic protein (MBP; red) and DAPI (blue; nuclei)-positive cells in external capsule (1A); corpus callosum (1B); cingulum (1C); and fimbriae of hippocampus (1D) from control (CS) and asphyxia-exposed (AS) rat neonates. Microphotographs show MBP, indicating both myelinated fibers and mature oligodendrocytes (OLs). Scale bar: 20 μm. At P1, no MBP immunoreactivity was observed in any of the analyzed regions and experimental conditions. The density of MBP increased significantly along development. At P7, the density of MBP fibers (white head arrows) was low, letting us visualize individual mature OL (white arrows). In corpus callosum and fimbriae of hippocampus some individual OL can also be seen, showing long and branched processes. In AS, there was a decrease in the density of MBP in external capsule, corpus callosum, and cingulum compared to that in CS. No differences could be seen in fimbriae of hippocampus at P7. At P14 a dense network of MBP fibers can be seen in all regions, but no independent OLs soma can be distinguished.

**Figure 2 ijms-22-03275-f002:**
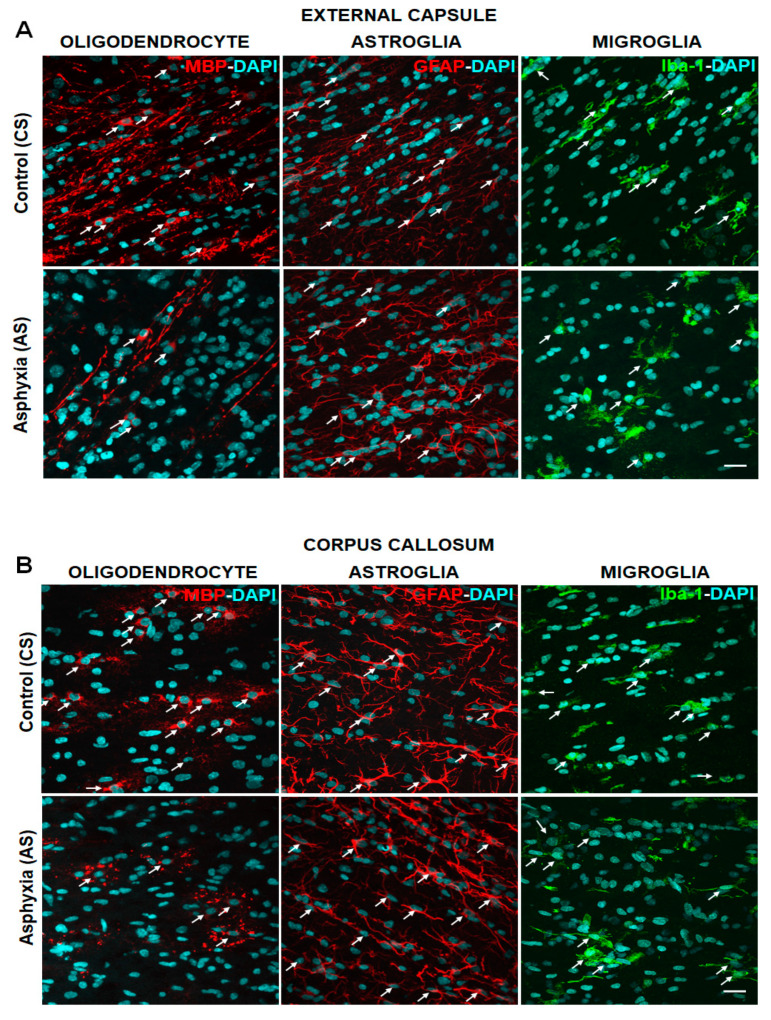
Effect of perinatal asphyxia (PA) on glial cells at P7, measured in external capsule (**A**); corpus callosum (**B**); cingulum (**C**); and fimbriae of hippocampus (**D**) of rat neonates. Representative microphotographs obtained by confocal microscopy showing myelin basic protein (MBP; red), glial fibrillary acidic protein (GFAP; red), ionized calcium binding adaptor molecule 1 (Iba-1; green) and DAPI (blue; nuclei)-positive cells in external capsule (2A); corpus callosum (2B); cingulum (2C); and fimbriae of hippocampus (2D), from control (CS) and asphyxia exposed (AS) rat neonates. White arrows show mature oligodendrocyte (OL), astrocyte, and microglia phenotype. Scale bar: 20 μm. At P7, the number of MBP-DAPI+ cells/mm^3^ decreased after PA in external capsule, corpus callosum, and cingulum, but not in fimbriae of the hippocampus.

**Figure 3 ijms-22-03275-f003:**
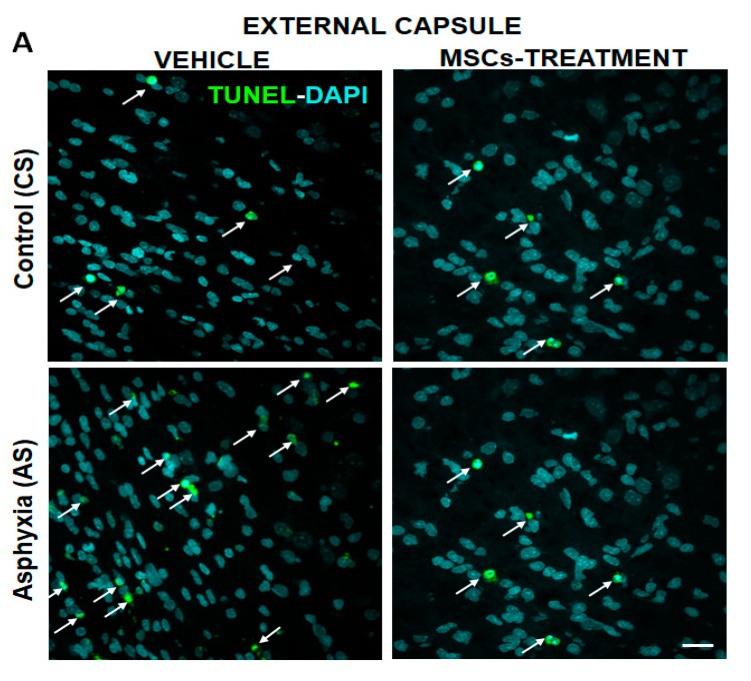
Effect of MSCs treatment on cell death induced by perinatal asphyxia (PA), measured at P7 in external capsule (**A**); corpus callosum (**B**); and cingulum (**C**) of rat neonates. Representative microphotographs obtained by confocal microscopy showing TUNEL (green), DAPI (nuclei, blue)-positive cells in external capsule (3A); corpus callosum (3B); and cingulum (3C) from vehicle- and MSCs-treated control (CS) and asphyxia-exposed (AS) neonates. Scale bar: 20 μm. (**A**–**C**). The number of TUNEL-DAPI cell/mm^3^ is increased when comparing vehicle-treated AS versus CS neonates, but the number of TUNEL-DAPI cell/mm^3^ is decreased in MSCs- versus vehicle-treated AS rat neonates in all evaluated regions.

**Figure 4 ijms-22-03275-f004:**
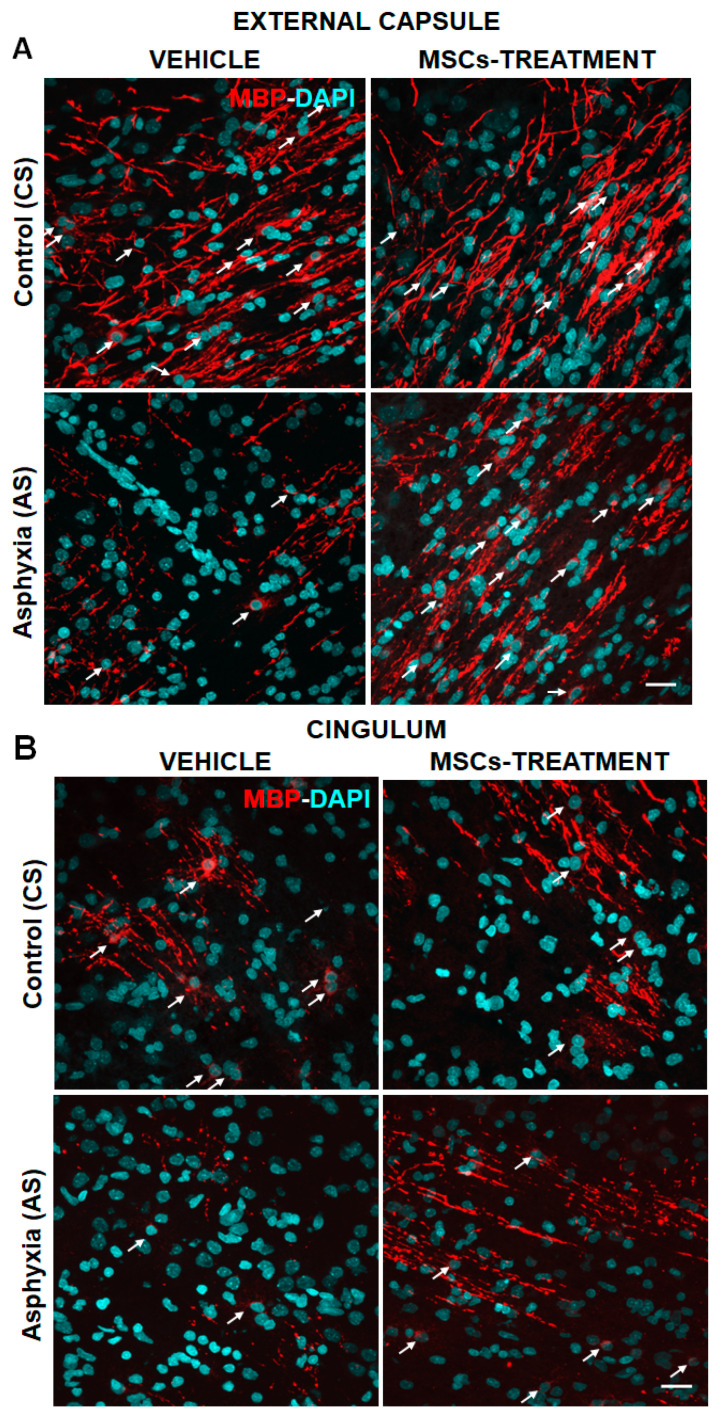
Effect of MSCs treatment on myelination and mature oligodendrocyte (OLs) injury induced by perinatal asphyxia (PA), measured at P7 in external capsule (**A**), and cingulum (**B**) of control (CS) and (AS) rats. Representative microphotographs obtained by confocal microscopy showing myelin basic protein (MBP; red) and DAPI (blue; nuclei)-positive cells in external capsule (4A); and cingulum (4B) from control (CS) and asphyxia-exposed (AS) rat neonates, including vehicle and MSCs treated groups. Microphotographs show MBP, indicating myelinated fibers (white head arrows) and mature oligodendrocytes (OL) (white arrows). Scale bar: 20 μm. The density of MBP and number of MBP-DAPI cells/mm^3^ is increased in MSCs- versus vehicle-treated AS neonates in all evaluated regions.

**Table 1 ijms-22-03275-t001:** Apgar and postnatal evaluation. The data are expressed as means ± SEM, from control (n = 75) and asphyxia-exposed rat neonates (n = 72). Whenever the parameters were monitored by continuous scales, or by % of the corresponding litter in cases of qualitative no continuous scales (n, number of pups; m, number of dams). The effect of neonatal development (**A**) and perinatal asphyxia (PA) (**B**) was evaluated on (i) survival (F(1, 45) = 7.701, *p* < 0.0001); (ii) body weight (F(6, 322) = 119.283, *p* < 0.0001); (iii) gasping (yes/no; in % for the corresponding litter) (F(4, 60) = 4.746, *p* < 0.002); (iv) respiratory frequency (events/min) (F(4, 178) = 107.689, *p* < 0.0001); (v) color of skin (% P, pink color skin, (F(4, 45) = 23.828, *p* < 0.0001); (vi) spontaneous movements (0–4; 0 = no movements; 4 = coordinated movements of forward and hind legs, as well as head and neck) (F(4, 138) = 98.523, *p* < 0.0001); (vii) vocalization % (yes/no; in % for the corresponding litter) (F(4, 70) = 10.440, *p* < 0.0001). (Benjamini–Hochberg as a post hoc test).

A. CS (n = 75; m = 20)	40 min	P1	P7	P14
Survival (%)	100	100	100	100
Body weight (g)	6.07 ± 0.08	6.60 ± 0.09	14.86 ± 0.93	23.05 ± 3.36
Gasping (%)	1.053 ± 1.05	0	0	0
Respiratory frequency (events/min)	74.29 ± 1.36	81.33 ± 3.77	94.67 ± 4.92	100 ± 5.59
Skin color (P%)	98 ± 1	100	100	100
Spontaneous Movements (0–4)	3.74 ± 0.07	4	4	4
Vocalizations (%)	98 ± 1	100	100	100
**B. AS (n = 72; m = 20)**				
Survival (%)	63 ± 6 **(by ~40%)b ******	100	100	100
Body weight (g)	7.38 ± 0.64	7.73 ± 0.91	13.90 ± 0.63	16.54 ± 2.49
Gasping (%)	**36.76 ± 10.51** **(> 35×)b ****	0	0	0
Respiratory frequency (events/min)	**32.53 ± 2.00** **(by ~40%)b ******	**59.00 ± 1.92** **(by ~20%)b ******	**75.20 ± 6.47** **(by ~20%)b ******	**86.00 ± 2.00** **(by ~14%)b ******
Skin color (P%)	0	100	100	100
Spontaneous Movements (0–4)	**0.50 ± 0.10** **(by ~90%)b ******	4	4	4
Vocalizations (%)	**44.72 ± 10.55** **(by ~55%)b ******	100	100	100

* *p* < 0.05, ** *p* < 0.01, *** *p* < 0.001 and **** *p* < 0.0001 (Bold).

**Table 2 ijms-22-03275-t002:** Effect of neonatal development and perinatal asphyxia (PA) on myelination (MBP; pixels/total pixels) in telencephalon (external capsule, corpus callosum, cingulum) and fimbriae of hippocampus) at P1, P7, and P14. Coronal sections of telencephalon and hippocampus were treated for immunohistochemistry against MBP, counterstained with DAPI. Microphotographs (two-five samples per brain region) were taken from external capsule, corpus callosum, cingulum, and fimbriae of hippocampus, in the field of a confocal-inverted Olympus-fv10i microscope at 60×. The mean percentage of MBP immunopositive pixels over the total pixel number was estimated using ImageJ software. Data are shown as means ± S.E.M from at least n = 5 independent experiments. Unbalanced two-way ANOVA indicated a significant effect of postnatal days (neonatal development) on MBP (pixels/total pixels) (**A**), increased in both CS (>2×, external capsule; >10×, corpus callosum; >6×, cingulum, and >25× in fimbriae of hippocampus) and AS (>5×, external capsule; >30×, corpus callosum; >20×, cingulum, and >30× in fimbriae of hippocampus) animals. At P7, MBP levels were decreased in external capsule, corpus callosum, and cingulum of AS versus CS animals (**B**) [external capsule (F(4, 25) = 230.066, *p* < 0.0001); corpus callosum (F(4, 25) = 3669.906, *p* < 0.0001); cingulum (F(4, 25) = 2037.145, *p* < 0.0001)], but not in fimbriae of hippocampus (F(4, 25) = 0.045, *p* = 0.833). Benjamini–Hochberg was used as a post hoc test. No statistically significant differences were observed in DAPI per mm^3^ in any of the evaluated regions.

Experimental Groups	P1		P7		P14	
A. Caesarean-Delivered (CS)	DAPI (cells/ mm^3^)	MBP (Pixels/Total Pixels)	DAPI (cells/ mm^3^)	MBP (Pixels/Total Pixels)	DAPI (cells/mm^3^)	MBP (Pixels/Total Pixels)
External capsule	374,444 ± 45,042	nd	355,067 ± 17,917	20.54 ± 1.14	367,240 ± 21,229	49.76 ± 0.12 **(>2×)a ******
Corpus callosum	262,787 ± 68,397	nd	235,916 ± 24,303	4.27 ± 1.03	325,644 ± 41,589	49.97 ± 0.15 **(>10×)a ******
Cingulum	378,496 ± 105,892	nd	300,722 ± 20,739	7.92 ± 0.82	388,988 ± 27,324	49.39 ± 0.33 **(>6×)a ******
Fimbriae of hippocampus	240,763 ± 67,331	nd	264,911 ± 19,963	1.94 ± 0.59	449,593 ± 51,446	49.97 ± 0.16 **(>25×)a ******
**B. Asphyxia-Exposed (AS); 21 Min Asphyxia**						
External capsule	521,962 ± 114,815	nd	392,315 ± 29,773	**9.88** **± 2.73** **(by 50%)b ***	398,727 ± 40,493	**49.67** **± 0.07** **(>5×)a ******
Corpus callosum	199,665 ± 55,743	nd	285,083 ± 19,983	**1.51** **± 0.47** **(by 60%)b ***	310,779 ± 37,973	**49.43** **± 0.14** **(>30×)a ******
Cingulum	438,835 ± 95,565	nd	346,825 ± 22,842	**2.29** **± 0.62** **(by~70%)b *****	411,625 ± 27,658	**49.61** **± 0.09 ** **(>20×)a ******
Fimbriae of hippocampus	269,789 ± 45,231	nd	286,527 ± 24,323	1.49 ± 0.62	344,219 ± 31,841	**48.65** **± 1.19** **(>30×)a ******

nd: not detectable.* *p* < 0.05, ** *p* < 0.01, *** *p* < 0.001, **** *p* < 0.0001 (Bold).

**Table 3 ijms-22-03275-t003:** Effect of neonatal development and perinatal asphyxia (PA) on mRNA levels of myelin basic protein (MBP), Oligodendrocyte transcription factors (Olig-1, Olig-2), and proinflammatory cytokines (IL-1β, IL-6, TNF-α, COX-2) evaluated in telencephalon from control (CS) and asphyxia-exposed (AS) rats at P1, P7, and P14. The effect of PA on MBP, Olig-1, Olig-2, and pro-inflammatory cytokines mRNA levels (a.u.) was determined by quantitative RT-qPCR analysis. Target genes were normalized by the housekeeping (β-actin), expressed as fold/change for each experimental group, compared to that evaluated at P1 in CS. Data are shown as means ± S.E.M., for independent experiments (at least n = 5). Unbalanced two-way ANOVA indicated a significant effect of postnatal days (**A**), and PA (**B**). MBP (F(4, 23) = 16.626, *p* < 0.0001); Olig-1 (F(4, 25) = 11.992, *p* < 0.0001); Olig2 (F(4, 26) = 5.573, *p* < 0.01); IL-1β (F(4, 24) = 11.539, *p* < 0.0001); IL-6 (F(4, 23) = 8.459, *p* < 0.0001); TNF-α (F(4, 22) = 2.335, *p* = 0.087); and COX-2 (F(4, 24) = 11.118, *p* <0.0001) mRNA expression. Benjamini–Hochberg was used as a post hoc test.

Gene Transcripts (mRNA) (Fold/Change)	P1	P7	P14
A. Caesarean Delivered (CS)	CS	CS	CS
MBP	1.000 ± 0.191	**23.23 ± 4.01** **(>20×)a ******	**1165 ± 258.90** **(>1000×)a ******
Olig-1	1.000 ± 0.098	1.37 ± 0.05	**2.315 ± 0.13** **(>2×)a ******
Olig-2	1.000 ± 0.076	1.24 ± 0.03	**1.826 ± 0.181** **(> 1.8×)a ******
IL-1β	1.000 ± 0.101	1.00 ± 0.13	**2.36 ± 0.18** **(>2×)a ****
IL-6	1.000 ± 0.201	0.70 ± 0.03	**1.86 ± 0.17 ** **(>1.8×)a *****
TNF-α	1.000 ± 0.208	2.14 ± 0.38	7.02 ± 3.63
Cox-2	1.000 ± 0.255	**5.84 ± 1.50** **(>5×)a ****	**33.56 ± 8.76** **(>30×)a ****
**B. Asphyxia-Exposed (AS)**	**AS**	**AS**	**AS**
MBP	0.91 ± 0.04	**35.63 ± 6.76** **(>35×)a ******	**768.90 ± 201.30** **(>800×)a *****
Olig-1	1.23 ± 0.05	**2.06 ± 0.36** **(>1.6×)a *** **(>1.4×)b ***	**2.40 ± 0.25** **(>1.9×)a ******
Olig-2	1.33 ± 0.15	1.61 ± 0.29	1.81 ± 0.21
IL-1β	1.50 ± 0.27	1.10 ± 0.08	2.46 ± 0.47
IL-6	0.78 ± 0.12	**1.16 ± 0.13** **(>1.6×)b ****	**1.46 ± 0.20** **(>1.8×)a ***
TNF-α	2.22 ± 0.67	8.32 ± 3.63	**40.42 ± 25.11** **(>18×)a ***
Cox-2	1.16 ± 0.34	**9.80 ± 1.59** **(>8×)a ******	**23.03 ± 6.00** **(>20×)a ****

* *p* < 0.05, ** *p* < 0.01, *** *p* < 0.001, **** *p* < 0.0001 (Bold).

**Table 4 ijms-22-03275-t004:** Effect of perinatal asphyxia (PA) on glial cells measured in telencephalon (external capsule, corpus callosum, cingulum) and fimbriae of hippocampus at P7. Control (CS) versus asphyxia-exposed (AS) rats, focusing on mature OLs (MBP), astrocytes (GFAP), and microglia (Iba-1). Coronal sections of telencephalon and hippocampus were treated for immunohistochemistry against MBP, GFAP, Iba-1, and counterstained with DAPI. Microphotographs (two-five samples per brain region) were taken from external capsule, corpus callosum, cingulum, and fimbriae of hippocampus, in the field of a confocal-inverted Olympus-fv10i microscope with a 60×. The mean number of the glial cells per mm^3^ was estimated using ImageJ software. Data are shown as means ± S.E.M from at least n = 5 independent experiments. Student’s *t*-test indicated a significant effect of PA on MBP-DAPI cells/mm^3^ in external capsule (t = 3.472, d.f. = 8, *p* < 0.008); corpus callosum (t = 4.143, d.f. = 8, *p* < 0.003); cingulum (t = 2.947, d.f. = 8, *p* < 0.019), but not in fimbriae of hippocampus (t = 1.462, d.f. = 8, *p* = 0.174). No significant differences were observed when comparing AS versus CS animals on GFAP-DAPI cells/mm^3^ in the external capsule (t = 0.564, d.f. = 8, *p* = 0.588); corpus callosum (t = 0.669, d.f. = 8, *p* = 0.523); cingulum (t = 1.696, d.f. = 8, *p* = 0.128), and fimbriae of hippocampus (t = 1.003, d.f. = 8, *p* = 0.345). Significant differences were observed when comparing AS versus CS animals on Iba-1-DAPI cells/mm^3^ in the cingulum (t = 2.329, d.f. = 8, *p* < 0.048), but not in external capsule (t = 0.793, d.f. = 8, *p* = 0.451); corpus callosum (t = 1.118, d.f. = 8, *p* = 0.296); and fimbriae of hippocampus (t = 0.042, d.f. = 8, *p* = 0.968).

Experimental Groups	P7			
A. Caesarean Delivered (CS)	DAPI cells/mm^3^	MBP-DAPI cells/mm^3^	GFAP-DAPI cells/mm^3^	Iba-1-DAPI cells/mm^3^
External capsule	346,895 ± 12,632	18,932 ± 1960	18,890 ± 1648	14,020 ± 1808
Corpus callosum	214,600 ± 14,080	10,810 ± 1668	19,090 ± 3229	8519 ± 561.6
Cingulum	294,902 ± 14,140	12,439 ± 1616	19,510 ± 3277	10,360 ± 955.6
Fimbriae of hippocampus	275,700 ± 15,350	8811 ± 2405	13,210 ± 2487	5957 ± 1027
**B. Asphyxia-Exposed (AS)**				
External capsule	370,346 ± 24,821	**8603 ± 982.7** **(by~50%)b ****	21,150 ± 3637	15,840 ± 1429
Corpus callosum	237,200 ± 20,170	**3413 ± 637.7** **(by~65%)b ****	21,280 ± 579.6	7380 ± 848.8
Cingulum	334,385 ± 18,199	**6643 ± 1121** **(by~45%)b ***	30,150 ± 5358	**17,800 ± 3047** **(>1.5×)b ***
Fimbriae of hippocampus	270,800 ± 19,810	4101 ± 2144	10,640 ± 591.70	6014 ± 909.1

* *p* < 0.05, ** *p* < 0.01, *** *p* < 0.001, **** *p* < 0.0001 (Bold).

**Table 5 ijms-22-03275-t005:** Effect of perinatal asphyxia (PA) on apoptotic-like (TUNEL-DAPI/mm^3^) and OLs-specific (TUNEL-MBP-DAPI/mm^3^) cell death in telencephalon from control (CS) and asphyxia-exposed (AS) rats at P7: Effect of neonatal MSCs treatment. Coronal sections of telencephalon were treated for immunohistochemistry against TUNEL, and MBP, counterstained with DAPI. Microphotographs (two-five samples per brain region) were taken from external capsule, corpus callosum, and cingulum, in the field of a confocal-inverted Olympus-fv10i microscope at 60×. The means of DAPI positive cells per mm^3^, alone; colocalizing with TUNEL/mm^3^, and MBP-TUNEL/mm^3^ were estimated using ImageJ software. Data are shown as means ± S.E.M from at least n = 5 independent experiments. No statistically significant difference was observed in DAPI per mm^3^ in any of the evaluated regions (external capsule F(2, 17) = 2.477, *p* = 0.114); corpus callosum (F(2, 17) = 3.279, *p* = 0.662); and cingulum (F(2, 17) = 1.257, *p* = 0.310). Unbalanced two-way ANOVA indicated a significant effect of PA on TUNEL-DAPI/mm^3^, increased in external capsule (>2.5), corpus callosum (>1.7×), and cingulum (>1.9×) of AS vehicle versus CS vehicle animals (a), but decreased significantly in external capsule (F(2, 17) = 10.430, *p* < 0.001); corpus callosum (F(2, 17) = 6.690, *p* < 0.007); and cingulum (F(2, 16) = 6.495, *p* < 0.009) of MSCs-treated compared to vehicle-treated AS rat neonates (b). TUNEL-MBP-DAPI+/mm^3^ levels were significantly increased in external capsule of AS vehicle (>6.5×) versus CS vehicle animals (a), but decreased significantly (F(2, 17) = 15.630, *p* < 0.0001) when MSCs-treated AS rat neonates were compared to vehicle-treated AS (b). Benjamini–Hochberg was used as a post hoc test.

Experimental Groups	P7		
A. CS-Vehicle	DAPI+/mm^3^	TUNEL-DAPI+/mm^3^	TUNEL-MBP-DAPI+/mm^3^
External capsule	382,789 ± 29,017	8145 ± 3543	152 ± 92.74
Corpus callosum	196,023 ± 12,124	5239 ± 1007	nd
Cingulum	346,834 ± 7511	7310 ± 2231	1138 ± 915.80
**B. CS-MSCs**	**P7**		
External capsule	378,613 ± 16,522	9198 ± 2064	277 ± 191.20
Corpus callosum	203,899 ± 7019	3975 ± 476	nd
Cingulum	379,610 ± 4365	4402 ± 1541	nd
**C. AS-Vehicle**	**P7**		
External capsule	341,872 ± 12,628	**22,837 ± 3732** **(> 2.5×)a *****	**1034 ±132.20** **(>6.5×)a ******
Corpus callosum	241,023 ± 21,694	**9276 ± 1942** **(>1.7×)a ***	261 ± 124.30
Cingulum	332,495 ± 28,634	**14,333 ± 2035** **(>1.9×)a ***	786 ± 423.90
**D. AS-MSCs**	**P7**		
External capsule	404,578 ± 21,398	**8145 ± 3543 ** **(by ~60%)b ****	**143 ± 89.23** **(by 85%)b ******
Corpus callosum	220,892 ± 8903	**4523 ± 642** **(by ~50%)b ***	nd
Cingulum	324,002 ± 44,070	**7147 ± 1790** **(by ~50%)b ****	nd

nd: not detectable. * *p* < 0.05, ** *p* < 0.01, *** *p* < 0.001, **** *p* < 0.0001 (Bold).

**Table 6 ijms-22-03275-t006:** Effect of perinatal asphyxia (PA) on oligodendrocytes (MBP-DAPI/mm^3^) and myelination (MBP positive pixels/total pixels) in telencephalon from control (CS) and asphyxia-exposed (AS) rats at P7: Effect of neonatal MSCs treatment. Coronal sections of telencephalon were treated for immunohistochemistry against myelin basic protein (MBP), counterstained with DAPI. Microphotographs (two-five samples per brain region) were taken from external capsule; corpus callosum, and cingulum, in the field of a confocal-inverted Olympus-fv10i microscope at 60×. The mean number of glial cells per mm^3^ and percentage of MBP immunopositive pixels over the total pixel number was estimated using ImageJ software. Data are shown as means ± S.E.M from at least n = 5 independent experiments. Unbalanced two-way ANOVA indicated a significant effect of PA on MBP-DAPI/mm^3^ and myelination (MBP pixels/total pixels), diminishing in external capsule and cingulum of AS vehicle versus CS vehicle (a), but increasing significantly in external capsule (F(2, 15) = 4.412, *p* < 0.03, DAPI/mm^3^; F(2, 16) = 2.813, *p* < 0.03, myelination, respectively); and cingulum (F(2, 13) = 2.632, *p* < 0.04, DAPI/mm^3^; F(2, 13) = 3.177, *p* < 0.03, myelination, respectively) of MSCs-compared to vehicle-treated AS rat neonates (b). No effect, however, was observed in corpus callosum (F(2, 15) = 0.738, *p* = 0.450, DAPI/mm^3^; F(2, 14) = 1.484, *p* = 0.151, myelination, respectively). Benjamini–Hochberg was used as a post hoc test.

Experimental Groups	P7	
A. CS-Vehicle	MBP-DAPI+/mm^3^	MBP (Pixels/Total Pixels)
External capsule	13,833 ± 1801	19.60 ± 1.32
Corpus callosum	6126 ± 1345	2.66 ± 0.44
Cingulum	9172 ± 2078	9.27 ± 1.61
**B. CS-MSCs**		
External capsule	13,632 ± 1783	16.36 ± 3.05
Corpus callosum	5972 ± 2775	2.24 ± 0.46
Cingulum	6572 ± 1585	8.91 ± 1.49
**C. AS-Vehicle**		
External capsule	6690 ± 2173 (by ~50%)a *	**13.08 ± 0.81** **(by ~30%)a ****
Corpus callosum	3942 ± 1255	1.54 ± 0.42
Cingulum	**3840 ± 509.30** **(by ~55%)a ***	**4.52 ± 0.10** **(by 40%)a ***
**D. AS-MSCs**		
External capsule	**13,628 ± 2237** **(>2×)b ***	**17.53 ± 1.70** **(>1.3×)b ***
Corpus callosum	7835 ± 2339	2.73 ± 0.62
Cingulum	**6194 ± 2145** **(>1.6×)b ***	**7.07 ± 1.47** **(>1.5×)b ***

* *p* < 0.05, ** *p* < 0.01, *** *p* < 0.001, **** *p* < 0.0001 (Bold).

## Data Availability

Not applicable.

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
