# Peer review of "Neonatal Mesenchymal Stem Cell Treatment Improves Myelination Impaired by Global Perinatal Asphyxia in Rats"

_ijms, 2021, doi:10.3390/ijms22063275_

Round 1

Reviewer 1 Report

In this article, entitled "Neonatal mesenchymal stem cell treatment improves myelination impaired by global perinatal asphyxia in rats", the authors describe the effects of MSCs treatment in perinatal asphyxia by using a rat model.

Although the authors suggested beneficial effects of the MSCs treatment, such as reduced cell death and enhanced myelination, this manuscript lacks of explanation how the MSCs regulate these effects. To improve the manuscript, the authors need to provide molecular or cellular mechanisms.

Author Response

Reviewer 1

In this article, entitled “Neonatal mesenchymal stem cell treatment improves myelination impaired by global perinatal asphyxia in rats”, the authors describe the effects of MSCs treatment in perinatal asphyxia by using a rat model.

Although the authors suggested beneficial effects of the MSCs treatment, such as reduced cell death and enhanced myelination, this manuscript lacks of explanation how the MSCs regulate these effects. To improve the manuscript, the authors need to provide molecular or cellular mechanisms”.

Reply

Thanks for your comments. We have expanded the discussion about mechanisms by which the effect of by MSCs can be explained. We are also referring to one of our recent published papers, also in IJMS (Farfan et al. 2020 Int J Mol Sci. 21(20):7800).

In previous studies, we demonstrated that perinatal asphyxia induces long lasting oxidative stress in the brain (Lespay-Rebolledo et al. Neurotoxicity, 2017, 2018). Furthermore, mesenchymal stem cells (MSCs) have been proposed as a therapeutic tool for neonatal central nervous system (CNS) diseases, due to their broad range of anti-inflammatory, anti-oxidant and pro-regenerative effects, involving both paracrine and cell-to-cell contact mechanisms. MSCs derived of adipose tissue are known to reduce oxidative stress (VallePrieto and Conget, 2010, Stem Cells Dev. 2010 19(12):1885-93) and to secrete anti-inflammatory cytokines including IL10 (Lee et al., 2016, Gut Liver 10:412–419.) and a soluble TNFα receptor, which neutralizes TNFα (Yagi et al., 2010, Ther 18:1857–64.).Indeed, the long-lasting antioxidant, anti-inflammatory and reparatory effects by MSC have been documented in adult animal models of stroke, ischemia, traumatic brain injury and chronic alcohol intake. Recent reports from our group showed that the administration of a single dose of adipose tissue-derived MSCs reduced continuous chronic alcohol intake by 50% during 10 days (Israel et al., Alcohol and Alcoholism, 2016). The levels of reactive oxygen species in hippocampus, as well astrocyte glial acidic fibrillary protein immunoreactivity, a hallmark of neuroinflammation induced by ethanol, which contributes to the perpetuation of ethanol intake, were markedly reduced by MSCs administration (Ezquer et al., Addict Biol. 2019 Jan;24(1):17-27). Similarly, intranasal administration of human-adipose MSC-derived secretome to rat neonates suffering PA, fully suppressed the long-lasting hippocampal oxidative stress (elevated GSSG/GSH ratio), promoting the nuclear translocation of Nuclear factor erythroid 2-related factor (Nrf2) triggering the production of various antioxidant enzymes and detoxification enzymes, including quinone oxidoreductases (NQO1).  The treatment also decreased nuclear NF-kB/p65 associated neuroinflammation, microglial reactivity, and cleaved-caspase-3 protein levels (a critical executioner of apoptosis) (Farfán et al 2020).  Whether the effect of a single administration of MSCs protects against the loss of mature oligodendrocytes and delayed myelination in telencephalic white matter via an antioxidant or an anti-inflammatory mechanism mediated by NRF2 activation remains to be determined.

Reviewer 2 Report

This study manuscript addressed the effect of perinatal asphyxia (PA) in rats on myelination and the glial cell response, and the potential for MSC treatment to improve these effects. This paper demonstrated that PA effected on myelination and OL maturation in external capsule, corpus callosum, cingulum and fimbriae of hippocampus. Then, MSCs treatment prevented the effect of PA on myelination and OL maturation in PA rat model. This study is well designed and conducted in appropriate methods. The paper provides the interesting data and this paper has merit for publication. Therefore, I think is entirely suitable for publication in the International Journal of Molecular Sciences. I have some minor suggestions to improve the quality of your manuscript as follows.

  1. Add the description that this study was performed using rat PA model in abstract.
  2. The introduction and discussion are redundant. If any statements are not relevant to this paper, please consider deleting them.
  3. Add the limitation of this study in the discussion.
  4. Add the month age of the donor rats for adipose derived mesenchymal stem cells.
  5. Did you confirm migration of MSCs administered into the ventricles to the parenchyma? In other words, do you speculate where did the cytokines, that improved the conditions of PA, secreted from MSCs? You need to discuss this.
  6. You should describe in more detail the effect of the discrepancy between IL-6 mRNA expression and protein expression on this experiment.

Author Response

Reviewer 2

This study manuscript addressed the effect of perinatal asphyxia (PA) in rats on myelination and the glial cell response, and the potential for MSC treatment to improve these effects. This paper demonstrated that PA effected on myelination and OL maturation in external capsule, corpus callosum, cingulum and fimbriae of hippocampus. Then, MSCs treatment prevented the effect of PA on myelination and OL maturation in PA rat model. This study is well designed and conducted in appropriate methods. The paper provides the interesting data and this paper has merit for publication. Therefore, I think is entirely suitable for publication in the International Journal of Molecular Sciences. I have some minor suggestions to improve the quality of your manuscript as follows”.

  • Add the description that this study was performed using rat PA model in abstract.”

Reply

We thank for the suggestion. The sentence “The issue has been investigated with a rat model of global PA” has been incorporated into the abstract.

  • The introduction and discussion are redundant. If any statements are not relevant to this paper, please consider deleting them.”

Reply

Thank you very much for the suggestion. We have extensively polished the manuscript, as it is shown in the labelled manuscript.

  • Add the limitation of this study in the discussion.”

Reply

Thank for the suggestion. Although the focus of this study was a first approach regarding the treatment with MSCs in neonates exposed to global PA, a model with a clinical correlate, further studies are certainly required to identify molecular mechanisms for the effects induced by MSCs on myelination. Also, the route of administration (i.c.v.) lacks clinical relevance. All these aspects have been incorporated into the discussion.

  • Add the month age of the donor rats for adipose-derived mesenchymal stem cells”.

Reply

Thank for the suggestion. The sentence “Wistar 12 week-old female rats (220-250 g)” was incorporated in methodology.

  • Did you confirm migration of MSCs administered into the ventricles to the parenchyma? In other words, do you speculate where did the cytokines, that improved the conditions of PA, secreted from MSCs? You need to discuss this”.

Reply

Thanks for your comments. We understand the opinion of the Reviewer. However, previous studies in our laboratory have evaluated the migration of bone marrow-derived and adipose tissue-derived MSCs, administered into the ventricles to the parenchyma (Israel et al. Intracerebral Stem Cell Administration Inhibits Relapse-like Alcohol Drinking in Rats. Alcohol and Alcoholism 2016). This reference was incorporate in discussion section.

  • You should describe in more detail the effect of the discrepancy between IL-6 mRNA expression and protein expression on this experiment”.

Reply

As reported in Table 3, IL-6 mRNA levels in asphyxia exposed-neonates increased only 1.6X with respect to control neonates, which does not necessarily imply an increase in protein levels. We conclude that IL-6 mRNA expression using RT-PCR to total RNA from brain samples is more sensitive than measuring directly protein levels. A similar effect was observed in our previous study (Neira-Pena et al. Perinatal asphyxia lead to PARP-1 overactivity, p65 translocation, IL-1β and TNF-α overexpression, and apoptotic-like cell death in mesencephalon of neonatal rats: prevention by systemic neonatal nicotinamide administration, Neurotoxicity Research 2015).

Round 2

Reviewer 1 Report

The novelty of the findings is somewhat of a concern. Although the authors did not show additional data to support the detailed mechanisms, they elaborated on potential regulations in the discussion. The authors need to interrogate molecular mechanisms in the future studies.